# Clinical and Electrophysiological Differences between Subjects with Dysphonetic Dyslexia and Non-Specific Reading Delay

**DOI:** 10.3390/brainsci8090172

**Published:** 2018-09-10

**Authors:** Jorge Bosch-Bayard, Valeria Peluso, Lidice Galan, Pedro Valdes Sosa, Giuseppe A. Chiarenza

**Affiliations:** 1Institute of Neurobiology, UNAM, Mexico City 76230, Mexico; oldgandalf@gmail.com; 2Centro Internazionale dei disturbi di apprendimento, attenzione e iperattività, (CIDAAI) Milano 20125, Italy; valeria.peluso@hotmail.it; 3Cuban Neuroscience Center Havana, Havana 10500, Cuba; lidicegalan2000@yahoo.com; 4MOE Key Lab for Neuroinformation, University of Electronic Science and Technology of China UESTC, Chengdu 610054, China; pedro.valdes.sosa@gmail.com

**Keywords:** direct test of reading and spelling (DTRS), dysphonetic dyslexia, non-specific reading delay, QEEG, quantitative EEG tomography, source localization, VARETA, Biomarkers

## Abstract

Reading is essentially a two-channel function, requiring the integration of intact visual and auditory processes both peripheral and central. It is essential for normal reading that these component processes go forward automatically. Based on this model, Boder described three main subtypes of dyslexia: dysphonetic dyslexia (DD), dyseidetic, mixed and besides a fourth group defined non-specific reading delay (NSRD). The subtypes are identified by an algorithm that considers the reading quotient and the % of errors in the spelling test. Chiarenza and Bindelli have developed the Direct Test of Reading and Spelling (DTRS), a computerized, modified and validated version to the Italian language of the Boder test. The sample consisted of 169 subjects with DD and 36 children with NSRD. The diagnosis of dyslexia was made according to the DSM-V criteria. The DTRS was used to identify the dyslexia subtypes and the NSRD group. 2–5 min of artefact-free EEG (electroencephalogram), recorded at rest with eyes closed, according to 10–20 system were analyzed. Stability based Biomarkers identification methodology was applied to the DTRS and the quantitative EEG (QEEG). The reading quotients and the errors of the reading and spelling test were significantly different in the two groups. The DD group had significantly higher activity in delta and theta bands compared to NSRD group in the frontal, central and parietal areas bilaterally. The classification equation for the QEEG, both at the scalp and the sources levels, obtained an area under the robust Receiver Operating Curve (ROC) of 0.73. However, we obtained a discrimination equation for the DTRS items which did not participate in the Boder classification algorithm, with a specificity and sensitivity of 0.94 to discriminate DD from NSRD. These results demonstrate for the first time the existence of different neuropsychological and neurophysiological patterns between children with DD and children with NSRD. They may also provide clinicians and therapists warning signals deriving from the anamnesis and the results of the DTRS that should lead to an earlier diagnosis of reading delay, which is usually very late diagnosed and therefore, untreated until the secondary school level.

## 1. Introduction

In 1973, Boder [1] claimed that reading is essentially a two-channel function, requiring the integration of intact visual and auditory processes both peripheral and central. It is essential for normal reading that these component processes go forward automatically. Based on this model, Boder and Jarrico [2] have developed a diagnostic screening procedure which identifies three main subtypes of dyslexia: dysphonetic dyslexia (DD), dyseidetic and mixed, besides a fourth group defined non-specific reading delay (NSRD). The identification of this further subgroup has evident clinical implications for diagnosis, therapy and prognosis. These subgroups are identified by an algorithm that considers the reading quotient and the percentage of errors of the spelling test. An Italian computerized version of the Boder test [2], the Direct Test of Reading and Spelling (DTRS) adapted and validated to Italian language, was developed by Chiarenza and Bindelli [3]. Luisi and Ruggerini [4] and Chiarenza and Di Pietro [5] using the DTRS, have shown that these subtypes of dyslexia and NSRD group exist also in Italian speakers and have almost the same prevalence as seen in dyslexics English-speaking subjects [2,6,7,8]. An indirect confirmation of the validity of Boder’s model has come from the study of information processing time through cognitive brain potentials. In dyslexic subjects, a disproportionate asynchrony exists between the auditory and visual system in the cross-modal processing of auditory and visual information [9,10].

EEG abnormalities in children with learning disabilities have been reported since long time. In a review paper, Chabot et al. [11] have summarized a variety of EEG abnormalities in individuals with dyslexia and related disorders, including poor EEG rhythm, low-voltage background rhythms [12] and increased generalized slowing [13]. However, the clinical utility of these findings has been questioned due to a lack of specificity [14]. Two subsequent studies provided more clinically relevant information [13,15]. EEG abnormalities were found in 24.6% of a sample of 61 children with learning disorders and in only 3.3% of control subjects. Abnormalities in children with learning disorders included increased high amplitude atypical alpha, abnormal focal paroxysmal activity, excess focal delta, persistent delta asymmetry, and excessive EEG response to hyperventilation. These abnormalities were most likely to occur in those LD children with significant pre- and/or postnatal risk factors for brain injury [15,16]. Thus, EEG studies indicate that specific developmental disorders are associated with abnormal EEGs in 25% to 43.5% of these children, with EEG slowing the most common abnormality reported.

Specific learning disabilities have been also extensively studied with new neuroimaging techniques: QEEG, functional Magnetic Resonance Imaging (fMRI), Single Photon Emission Computed Tomography (SPECT), and Diffusion Tensor Imaging (DTI). A variety of structural and functional connectivity abnormalities in patients with dyslexia has been identified [17,18,19]. Using DTI it has been reported that the degree of myelination in left hemisphere cortico-cortical tracts correlates positively with reading skill [20,21] and reduced myelination in left hemisphere white matter tracts connecting inferior frontal, temporal, occipital, and parietal regions has been described among adults with a history of reading disability [22]. Both increased and decreased anatomical using DTI and functional connectivity (task-related fMRI) within the reading network of dorsal and ventral brain regions have been reported in adults with reading difficulties compared to normal readers [23,24,25,26].

Since the pioneer work of John et al. [27] and Duffy et al. [28], QEEG abnormalities in both hemispheres have been reported in dyslexic children at rest as well as during complex testing. John (1981) [29] found abnormal QEEGs present in 32.7% of the children with Specific Learning Disorder (SLD) and 38.1% of the children with learning disorder (LD) groups, whereas only 5.5% of an independent sample of normal children had abnormal QEEGs. Discriminant analyses using six multivariate features resulted in 80% correct classification of normal children (87% replication) and 72% correct classification of LD children (65% replication). Using similar neurometric techniques, Marosi et al. [30,31], John and Prichep [32] have further elucidated the nature of neurophysiological abnormality in children with documented learning disorders. Children with learning disorders were shown to have different patterns of brain maturation than normal control subjects. Normal subjects, with increased age show an increase of posterior/vertex EEG coherence and a decrease in coherence in frontal areas. Increased differentiation of frontal cortical regions, with age, leads to increased communication across basic sensory and association cortex. These systematic maturational changes are often not seen in children with learning disorders. Instead, they show no change in posterior/vertex coherence with age, and levels of frontal coherence remain high across all ages, suggesting a deviation from normal development rather than a maturational lag [30,33]. More recently, abnormally high activity in the delta and theta band has been reported in a group of children with learning disabilities not otherwise specified (LD-NOS) who had low scores in reading, writing and calculation tests [34,35]. Flynn and Deering [36] and Flynn et al. [37], using the Boder test classification [2], investigated whether electrophysiological evidence among dyslexic subgroups, could be demonstrated by analysis of QEEG patterns during school-related tasks. The authors found increased left temporo-parietal theta activity in dyseidetic children assuming an overuse of the left-language system activation in patients with visuo-spatial problems recognition.

Other neuroimaging studies using reading-related potentials recorded during reading aloud self-paced single-letter showed that children with DD had abnormal activation at short latencies in the left temporal polar area, at middle latencies in the temporal polar and inferior frontal regions bilaterally and at long latencies in fronto-temporal regions of the right hemisphere [38]. This indicates an early involvement of frontal regions during reading and it may be related to a significantly higher activation of the right hemisphere. This would seem very likely to be related to compensatory mechanisms adopted by reading-impaired children to improve their performance. On the contrary, impaired neural activation of the dyslexic group was present in the left and medial parietal regions: at short and middle latencies, it was present in the angular and then in the supramarginal gyrus; at long latencies, it moved in the middle precuneus and occipital lobe. Therefore, behavioral signs of reading impairment can be related to reduced activation in the left dorsal parieto-occipital regions that have been shown to be specifically involved in reading processes and particularly in the storage and processing of the visual and auditory representations of alphabetic characters [39]. These results are consistent with previous findings of greater recruitment of cerebral regions in the right hemisphere in dyslexic children in comparison with controls [40].

Although all the cited studies identify the left temporo-parietal region as the region where the greatest differences between normal and dyslexic subjects are found, electrophysiological differences between subtypes of dyslexia according to Boder classification have not been further confirmed also due to different clinical classifications. Furthermore, we are not aware of studies that compare children with different subtypes of dyslexia with children with NSRD using electrophysiological methods.

QEEG can be used to indicate which children with learning problems have a measurable underlying neurophysiological dysfunction and which do not. The purpose of this research was to find differences between DD and NSRD groups, both at the neurophysiological and neuropsychological levels. At the same time, we wanted to find a classification equation which discriminates the two groups with a high percent of accuracy. This could contribute to the earlier diagnosis of the NSRD group, usually very late diagnosed and therefore, untreated until the secondary school level.

## 2. Materials and Methods

### 2.1. Subjects (Entire Sample)

Seven-hundred-seventy-onesubjects, 488 males between the ages of 7 and 16 (mean age 9.24; SD 0.9) and 283 females between the ages of 6 and 18 (mean age 8.99.2; SD 1.98) were examined at our center for suspected developmental dyslexia. After administration of the DTRS, 83 subjects were diagnosed as normal (10.7%), 113 subjects had NSRD (14.6%), 394 had DD (51.1%), 43 had dyseidetic dyslexia (5.6%), 107 subjects had mixed dyslexia (13.9%), 31 subjects were remediated (4.0%).

#### Composition of the Subsample

One-hundred-sixty-nine out of the 394 DD subjects had EEG studies: 13 of them had 2 studies and 2 of them had 3 studies. From the NSRD, 36 out of 113 had EEG studies: 8 of them with 2 studies. Statistical analysis was then performed on 169 DD and 36 subjects with NSRD. Due to the big differences between number of subjects in both groups, a special version of the t-test for non-balanced samples was used.

The gender composition of the EEG subsample was as follows: DD: 110 males between the ages of 7 and 15 (mean age 9.4; SD 1.9) and 59 females between the ages of 7 and 14 (mean age 8.8; SD 1.8); NSRD: 22 males between the ages of 7 and 14 (mean age 9.9; SD 1.9) and 14 females between the ages of 7 and 15 (mean age 9.6; SD 2.4).

### 2.2. Clinical Protocol

The clinical assessment was conducted according to the following protocol: the parents (one or both) were clinically interviewed to obtain family history and the following information was obtained: demographics, parents qualification, medical and psychiatric history including the presence of language delay or specific language disorders, previous and concomitant medications and the eventual type of speech therapy performed or still in progress and its duration. At the first visit, the informed consent was also obtained from parents/caretaker, adolescents and children after explaining them the purpose and the procedures of the study. Physical and neurological examination, EEG, Amsterdam Neuropsychological Test (ANT) [41], a battery to test executive functions and attention, were carried out in the following visits.

The diagnosis of dyslexia was carried out according to Diagnostic and Statistical Manual of Mental Disorders (DSM-V) [42] criteria with the administration of the following tests: (a) WISC III, (b) reading tests for primary school [43], and for secondary school [44] providing accuracy and speed scores in reading aloud age-normed texts, (c) single word and non-word reading, also providing speed and accuracy scores for each grade: the battery for the assessment of dyslexia and developmental dysortography [45], and (d) the Direct test of Reading and Spelling [3,46]. Test AC-MT 6-11: Test di Valutazione Delle Abilità di Calcolo (AC-MT), a battery for dyscalculia [47,48] or the battery for the assessment of developmental dyscalculia [49] were carried out if the anamnesis indicates difficulties in mathematics. The presence of dysgraphia was assessed with Scala Sintetica per la Valutazione Della Scrittura in Età Evolutiva (BHK) [50] and Batteria per la Valutazione della Scrittura e della Competenza Ortografica (BVSCO) tests [51].

In 48 subjects out of 169, dyslexia was associated with dysortography (28.4%), in 6 subjects with dysgraphia (3.5%), in 8 subjects with dyscalculia (4.7%), and in 14 subjects with dysortography, dysgraphia and dyscalculia (8.3%). Eight DD subjects had a previous diagnosis of Specific Language Disorder (SLD). Of these 8 subjects, five DD children received speech therapy lasting 12–48 months. Of the group of 36 subjects with NSRD, 11 subjects were dysortographic (30.5%), 2 subjects dysgraphic (5.5%), 3 subjects had dyscalculia (8.73%) and 4 subjects had dysortography, dysgraphia and dyscalculia (11.1%). One subject with NSRD had a previous diagnosis of Specific Language Disorder (SLD). No therapy had been prescribed for this subject with NSRD.

The most frequent comorbidity in both groups was Attention Deficit and Hyperactivity Disorder (ADHD). In the DD group, 34 subjects (20.1% of total DD subjects) were affected by ADHD: 16 subjects (47.0%) had ADHD of inattentive subtype, 18 subjects (52.9%) had ADHD of combined subtype. In the group of subjects with NSRD, 12 subjects were affected by ADHD (33.3% of the total subjects with NSRD): 6 subjects had ADHD of inattentive type (50%), 5 subjects had ADHD combined subtype (41.7%) and 1 subject with ADHD not specified (8.3%).

The 169 DD subjects had a mean Full Scale Intelligent Quotient (FSIQ) of 101.4 (SD: 10.9), a mean Verbal Intelligent Quotient (VIQ) of 99.6 (SD: 11.7) and a mean Performance Intelligent Quotient (PIQ) of 103.4 (SD: 12.1). The 36 subjects with NSRD had a mean FSIQ of (105.5(SD: 9.2), mean VIQ of 103.9 (SD: 12.1) and a mean PIQ of 106.6(SD: 10.0).

All subjects had neurological examination within normal limits.

#### Exclusion Criteria

Presence of documented psychiatric disorders in parents, a documented history of Bipolar disorders, history of psychosis or pervasive developmental disorder, seizure disorder, head injury with loss of consciousness or concussion, migraine, neurological/systemic medical disease (e.g., lupus, diabetes) or history of stroke or arterious-venus malformation or brain surgery were considered an exclusion criterion. Functional comorbidities such as visual or auditory processing problems were documented with IQ testing. The presence of ADHD or mild anxiety disorders were not exclusion criteria.

### 2.3. The Direct Test Reading and Spelling (DTRS)

The DTRS is self-administered and self-paced [3,46]. The DTRS consists of a reading and spelling test. The reading test has 15 lists. The first four are for the first grade of the primary school; the other 10, two for each grade, for the other five grades and the last one for the sixth grade. Each list consists of 20 words, ordered by increasing difficulty according to numbers of syllables and orthographic difficulties (following this design, a complexity index was constructed, which is explained and used in the statistical section). The subject decides spontaneously by pressing a button with the dominant hand to display on a screen the word to be read aloud. A microphone records the subject’s phonogram used to measure the reading time of each word. The words lists are presented in two ways: ‘flash’ and ‘untimed’ mode. In ‘flash mode’ the word appears for 250 ms, which determines the child’s *sight vocabulary* (i.e., the words the child recognizes instantly as whole word configuration or gestalts). If he/she recognizes a word within one second, this is recorded as ‘word flash correct’ (FCOR). If he/she hesitates (i.e.: hem ... hem ... ‘ice-cream’) or reads the word syllable by syllable, it is classified as ‘flash hesitating’ (FHESIT) and ‘flash syllabicating’ (FSILL) respectively. If he misreads the word or does not read it at all the child is asked to try again and, the word appears for 10 s ‘untimed mode’, which calls upon the child’s ability to analyze unfamiliar words phonetically (i.e., his *word analysis-synthesis skills)*. If he identifies the word correctly within 10 s this is recorded as ‘word untimed correct’ (UCOR). If he misreads the word or reads the word with great difficulty they are recorded as ‘word untimed incorrectly read’ (UINCOR) or words read with great difficulty (UGDIFF). The words not read are recorded as ‘NR’.

Comparison of the number of correctly read words in the ‘flash’ or ‘untimed’ mode indicates whether the child is reading through both whole-word gestalt and phonetic analysis or predominantly trough one or other. The highest-grade level at which the child’s sight vocabulary includes at least 50 per cent of the word list is considered his reading level (RL). Misreadings are recorded for later evaluation of the child’s characteristic errors of the subtype of dyslexia. The description of these typical errors of dysphonetic and dyseidetic dyslexia is reported in [Sec secA1-brainsci-08-00172].

The spelling test is complementary to the reading test. It consists of dictating to the subject two lists of ten words each: a list of known words (KW) chosen from the ‘sight vocabulary’, and a list of unknown words (UW), chosen from those unread or read with great difficulty during ‘untimed’ mode (i.e., not in sight vocabulary). The examiner notes the number of correctly spelled words in both lists and the type of errors. The description of the spelling errors is reported in [Sec secA2-brainsci-08-00172]. Analysis of the spelling of ‘known words’ reveals the child’s ability to ‘revisualize’ the words in his sight vocabulary, and analysis of his list of ‘unknown words’ reveals his ability to spell words not in his sight vocabulary. Thus, the two spelling lists are designed to tap the central visual and auditory processes necessary for spelling, in the same way that the ‘flash’ and ‘untimed mode’ of the reading test tap the central visual and auditory processes necessary for reading.

Two independent researchers conducted both a qualitative and quantitative analysis of the errors of the reading and spelling test of the subjects of the entire sample according to the classification proposed by Boder [1] with the addition of some errors typical of the Italian language as accents and double letters.

At the end of the reading test the computer automatically provides the reading level (‘RL’), the reading age (‘RA’), and the reading quotient (‘RQ’): the ratio between ‘RA’ and chronological age (CA) (RQ = (RA/CA) × 100). If the child’s overall mental ability is substantially above or below average, this quotient is corrected for mental age (MA) by use of the following formula, RQM = (RA/MA) × 100. The RQC is the reading quotient corrected for mental and chronological age according to the following formula, RQC = (2RA/(MA + CA) × 100.

The computer provides also the number of words read correctly in flash (FCOR) and untimed (UCOR) mode and the average reading time of the words of each list read correctly in ‘flash mode’’ (RTFCOR) and in ‘untimed mode’ (RTUCOR), the average reading time of the words of each list read after marked hesitation (RTFHESIT), the average reading time of the words of each list read incorrectly (RTINCOR) the average reading time of the words of each list read with great difficulty after one or two attempts (RTUGDIFF (the word is read only partially or with the addition of letters or syllables, or with wrong accent); Reading time consists in the time that elapses between the occurrence of the word on the screen and the start of the pronunciation of the word recorded through the phonogram.

Finally, the identification of the dysphonetic reading-spelling pattern and that one of unspecific reading delay is based on the child’s performance in the three basic diagnostic indicators: % of words spelled correctly in the known list, % of words spelled correctly in the unknown list and the reading quotient. The dysphonetic pattern is present when the child has a percentage of words correctly spelled in the two lists less than 70% and a reading quotient greater than 67; the non-specific reading delay pattern is present when the child has a percentage of words correctly spelled in the two lists greater than 70 and a reading quotient of less than 90.

#### The Complexity Index

To obtain a deeper understanding about the reading differences between the DD and NSRD, the reading time results are evaluated. Since children read different lists and words depending on their age and grade, to make them comparable, a complexity index was created. Each word in the list was divided in % of complexity according to the number of syllables and number of orthographic difficulties. Since each list has 20 words, for all individuals 20 levels of degree of complexity for both flash and untimed mode were created. For each level of this normalized index, the corresponding reading time was considered and both groups were compared.

### 2.4. Neurophysiologic Assessment

#### 2.4.1. EEG Data Acquisition

The EEG was recorded at 19 leads of the 1020 International Positioning System (S10–20), using Electro-caps referenced to linked earlobes. Twenty minutes of eyes closed resting EEG were recorded. A differential eye channel (diagonally placed above and below the eye orbit) was used for detection of eye movements. All electrode impedances were below 5000 Ohms. The EEG amplifiers were set to a bandpass from 0.5 to 70 Hz (3 dB points). All EEG data were collected on the same digital system to achieve amplifier equivalence. Data were sampled at a rate of 256 Hz with 12-bit resolution. All the patients were recorded in the morning and instructed to keep their eyes closed and stay awake. This allowed to control for drowsiness during EEG recordings and to guarantee similar conditions throughout the different sessions. The technician was also aware of the subject’s state to avoid drowsiness. Additionally, patients were monitored with a closed-circuit television system, during the recording.

#### 2.4.2. QEEG at the Scalp

EEG experts visually edited the raw EEG data to select EEG epochs free of either biological (e.g., muscle movement, EMG) as well as non-biological (e.g., electrical noise in the room) artifacts. This was augmented by a computerized artifact detection algorithm. A minimum of 24 epochs of 2.56 minutes of artifact-free data, were selected for each subject and submitted to frequency analysis. The EEG spectra were calculated using the High Resolution Spectral (HRS) model [52,53,54] for all the channels by means of the Fast Fourier Transform (FFT), in a frequency range from 0.39 Hz to 19.11 Hz, with a frequency resolution of 0.39 Hz. The selection of these frequency parameters was made on the basis that they match the available parameters from the Cuban Normative Database [53] to transform the raw EEG spectra into Z-probabilistic measurements, age corrected. The spectra were Log transformed, to approach them to Gaussianity [55,56] and the Z-transform was calculated against the Cuban Normative Database. Significant test-retest reliability for these measures has been demonstrated [27,57,58].

#### 2.4.3. QEEG Source Analysis (QEEGT)

To obtain the raw spectra at the EEG generators, the Variable Resolution Electrical Tomography (VARETA) method [59] was used for the source localization analysis. Same as with the spectra at the electrodes, the source density localization analysis was performed for frequencies between 0.39 to 19.11 Hz and for all the sources in the cerebral cortex for a grid of 3244 sources. VARETA is an already known technique for estimating the distribution of the primary current in the source generators of EEG data. VARETA is a Discrete Spline Distributed Solution, like LORETA [52]. The spline estimates are the spatially-smoothest solutions compatible with the observed data. VARETA however, adapts to the actual degree of smoothness in each voxel being determined by the data itself, by applying different amounts of spatial smoothing for different types of generators; hence the use of the term variable resolution. Additionally, VARETA allows spatially-adaptive nonlinear estimates of current sources eliminating ‘ghost solutions’ (artefactual interference patterns) often present in linearly-distributed inverse solutions. Therefore, VARETA produces focal solutions for point sources, as well as distributed solutions for diffuse sources. In addition, VARETA introduced the use of anatomical constraints upon the allowable solutions by introducing a ‘gray matter weight’ for each voxel. The effect of these weights is to prohibit sources in voxels where the mask is zero (for example, CSF or white matter). To solve the EEG forward problem, a three-concentric spheres Lead Field [60], defined over a grid of 3244 points located in the gray matter of the Montreal Neurological Institute (MNI) template [61] was used to generate the voltage at the 19 electrodes of the 10–20 System.

The average reference was applied to the EEG at each time point for this analysis, as required by VARETA and other inverse solution methods. To make the inverse solution in different subjects comparable for statistics at the sources, the same regularization parameter for all subjects was provided to VARETA, which was obtained from the normative database, and it was also used for the norms calculations. The geometric power was applied to standardize the EEGs by a global scale factor. This factor accounts for individual differences in power values due to skull thickness, hair volume, electrode impedance, and other factors of variance that can affect EEG amplitudes not related to electrophysiological variability [62]. This procedure is described in detail in Bosch-Bayard et al. [59]. The anatomical accuracy of the functional QEEG source localization used in VARETA has been repeatedly confirmed by co-registration with other brain imaging modalities e.g., fMRI [63], positron emission tomography, PET [64,65], and computerized tomography [66]. It has also been used satisfactorily for source localization in psychiatric patients [67].

The selected EEG epochs were submitted to VARETA for the computation of the spectra at the EEG sources, using the 3244 sources provided by the grid defined over the gray matter of the probabilistic MRI Brain Atlas [61,68]. The Lead Field was calculated according to [60], using the standard positions of the International 1020 electrodes positioning system. Using this Lead Field, VARETA was applied to the cross-spectral matrices at the scalp for all frequencies and leads, by applying the methodology described in Bosch-Bayard et al. [59] to calculate the spectra at the sources space with the same frequency resolution (from 0.39 to 20 Hz with a frequency resolution of 0.39 Hz: high spectral resolution, HSR). The procedure of obtaining the spectra for every source of the gray matter at each frequency constitutes a new type of neuroimaging technique that has been termed Quantitative EEG Tomographic Analysis (QEEGT) [59].

As pointed out above, to account for age differences, the Z-spectra were computed relative to the normative values of the Cuban Normative Database. The values computed for the 3244 sources were encoded using a color palette with hues proportional to the standard- or Z-scores of deviations from expected normative values. The random fields theory [69] was used to correct the significance levels of the images for the large number of comparisons. This approach proposes a unified statistical theory for assessing the significance of apparent signal observed in noisy difference images, giving an estimate of the corrected p-value for local maxima of Gaussian fields over search regions of any shape or size in any number of dimensions.

#### 2.4.4. Stability Based Biomarkers Identification

The methodology used in this paper has been described in detail in Bosch-Bayard et al. [34]. The algorithm is an approach for solving the classification problem that guarantees stability and robustness, a property of great usefulness in problems where the sample is small, and the number of variables is high. The method is divided in two steps, one for the parameter estimation and the other for the validation procedure. In the first step, a random subsample of the original sample is generated using the 30% of the variables and the 70% of the subjects and the classification procedure is applied to this subsample. We use the General Linear Model via elastic net (GLMNet) [70] which is a logistic penalized regression method. This method extracts a sparse subset of classifiers from the whole set of variables, while discarding the rest. The random procedure is repeated a sufficiently number of times (500 times in our case), storing the classifiers in each of them. The variables selected as classifiers in at least 50% of the times that they participated in the classification procedure integrate the final set of biomarkers. To measure the classification power of the selected set of biomarkers, a robust version of the ROC technique is used. In this case, all biomarkers are kept while generating a subsample with about the 70% of the subjects. The ROC area is calculated for the 10, 20 and 100% of False Positive (FP) and these numbers are stored. The procedure is randomly repeated a sufficiently high number of times (500 in our case). Three empirical distributions of the ROC values are obtained for the two levels of FP and the total area. The values of these distributions at the 50% of the curve are given as the stable estimate of area under the ROC, which characterizes the predictive classification power of the selected set of biomarkers.

### 2.5. Statistical Analyses

Group differences were tested for the DTRS variables as well as the QEEG spectra at the scalp and at the sources by means of the t-Student. To account for the very unbalanced samples, a special algorithm was developed, which also corrected the levels of significance for multi comparisons. The algorithm consisted in generating 100 random subsamples of the DD group, of the same length as the NSRD group and performing t-tests between them. For each repetition the t and the *P* values were stored, and empirical distributions of the *t* values were created for all the variables and subsamples. The corrected values at the 0.05 threshold were obtained by taken the *t* values at the 5% and 95% of the empirical distribution.

Additionally, a classification algorithm for biomarkers detection between the two groups was applied to each dataset [34]: DTRS and the QEEGT spectra at the scalp and at the sources. Since the DTRS is the instrument used in the Boder algorithm to classify the subjects into the DD and NSRD groups, we only included in the biomarkers selection procedure those DTRS items which were not used in the Boder algorithm (% of known words, % of unknown words, RL, RA, RQ, RQM, RQC). Therefore, in the case of the DTRS data, we found a set of biomarkers completely independent to those used for the creation of the dyslexia subgroups.

## 3. Results

### 3.1. Results of the DTRS

From the collection of anamnestic and clinical data, the comparison between the two groups of subjects revealed that the age at the time of DTRS testing was highly significant (*p* = 0.0129, *t* = −2.51). The NSRD were significantly older at the time of testing than the DD children. Sex, FSIQ, VIQ and PIQ, the presence of specific language disorder, the associated diagnosis of dysorthography, dysgraphia, and dyscalculia and parental schooling were not significantly different in the two groups. Also, the presence of speech therapy, the type of speech therapy and its duration was not significantly different between the two groups. Subjects with NSRD had a greater presence of comorbidity than DD subjects, but the type of comorbidity was not significantly different in the two groups (Table 1).

Table 2 shows the results of the reading test of DTRS in the two groups. As well as age at the time of DTRS administration also the school level was significantly different in the two groups. Subjects with NSRD were on a higher school level than DD subjects at the time of testing. The number of lists read by the subjects, the reading level, i.e., the last list where the subjects read less than 50% of the words in flash mode and the reading age were not significantly different in the two groups. The subjects with NSRD had a significant lower RQ, RQM and RQC than the subjects with DD.

Table 3 shows the results of the spelling test of DTRS in the two groups. The percentage of known words and unknown words spelled correctly was significantly higher in the NSRD group compared to the DD children. The qualitative and quantitative analysis of the errors of the spelling test showed that the insertion/omission of letters was among the various types of errors (see [Sec secA2-brainsci-08-00172]) that were significantly different between the two groups. Insertion/omission of letters were significantly higher in DD subjects than subjects with NSRD. In addition, the DD subjects had a total number of errors of dysphonetic and dyseidetic type significantly greater than the NSRD subjects in the list of known words correctly spelled. The DD subjects made a greater number of errors of dysphonetic type compared to the NSRD subjects in the list of unknown words. The most frequent errors made by the DD subjects were accentuation errors, lack of double, insertion/omission of letters, difficulty in the grapheme-phoneme correspondence and grapheme exchange.

Table 4 shows the results of the comparison of the number of words read correctly and incorrectly in flash and untimed mode and their relative reading times between the two groups. The number of words read correctly and incorrectly and their relative reading times in both flash and untimed mode were not significantly different between the two groups. However, it is interesting to report that the only significant difference between the two groups was found in the number of words read in a hesitant way. The NSRD subjects had a significantly greater number of words read in this mode than the DD subjects.

Table 5 shows the t test results of reading times of DD and NSRD groups according to the complexity index by syllables and number of difficulties and by reading mode (flash, untimed).

The most consistent and significant results were found in the reading time of words correctly read and read with evident difficulty in untimed mode. The DD subjects read more slowly than subjects with NSRD when they faced words with a greater number of syllables that also contained a greater number of orthographic difficulties. The greatest differences were found with words of 4 or 5 syllables; however with words of 6 or 7 syllables there were no significant differences between the groups.

### 3.2. Results of the QEEG

The t-tests algorithm described in Section 2.5 was used to compare the EEG Z-spectra of the two groups for all leads and frequencies of the High Spectra Resolution (HSR) model. The Z-spectra values have been corrected by the age and account for EEG differences due to maturation since this is an important factor to be considered at this epoch of the life. The results are shown in Figure 1. In this comparison DD group was subtracted from the NSRD group, therefore patches in red in Figure 1 represent excess of activity in DD group regarding NSRD group and the blue patches represent leads and frequencies where the activity of DD group was significantly lower than NSRD. As it is observed, the DD children showed significant excess in delta band in the middle line (Fz, Cz and Pz), as well as Fp2 and the occipital leads bilaterally (O1 and O2). A significant excess in high theta (6–7.5 Hz) and low alpha (7.5–8.5 Hz) bands was also present in the Fz, Cz and Fp2. Fz, Cz and Pz also showed significant excess of activity in the DD group. However, a significant reduction of high alpha (11–12.5 Hz) activity was present in the DD group bilaterally in F3, C3, C4 and in P3 but more pronounced in the left leads. Additionally, significant reduction was also present in the left leads F7, F3, C3, P3 and T5.

The *t*-tests at the sources showed a significant increase of activity of DD with respect to NSRD in Delta, low (4.29 Hz) and high theta (7.5 Hz) bands, and a significant decrease of DD with respect to NSRD in Beta band (18–19 Hz). In Delta band: bilaterally in the Calcarine sulcus, Cuneus, Precuneus, Lingual, Occipital (Superior, Middle and Inferior lobes), Fusiform and Superior Parietal; the Right Inferior Parietal, the right angular gyrus and the right Paracentral Lobule. In the low theta band: the right superior parietal and the right inferior parietal Gyrus (or lobe). In the high theta band: bilaterally the Frontal Medial Orbital and the Right Superior Medial Frontal gyri. In the Beta band:bilaterally the Calcarine sulcus, Lingual and Fusiform Inferior gyri; the Left Occipital (Superior, Medial and Inferior), Superior and Inferior Parietal, Supramarginal, Angular and Middle and Inferior Temporal (Figure 2).

### 3.3. Biomarkers Results

We applied the biomarkers selection methodology to the variables of the DTRS as well as to the Z-spectra at the electrodes and at the sources separately. The discrimination power of each classification equation was measured by the robust area under the ROC curve (rAUC) and the ROC discrimination value at the 10% and 20% of False Positives.

Table 6 summarizes the variables of the anamnestic data and of DTRS selected as classifiers during the classification procedure. The second column indicates the percent of times that the variable was selected as biomarker during the random classification procedure. The last column shows the coefficient of each variable in the final classification equation. Class, comorbidity and speech therapy were selected from the anamnestic data, as the most frequent ones (more than 95%). The FSIQ, the VIQ, the age and the school attended at the time of DTRS testing also participated in the classification procedure. The variables of the DTRS that contribute the most to the classification procedure were the dysphonetic errors in the known words list. The errors that best contributed to differentiate the two groups were: the grapheme-phoneme correspondence, the insertion/omission of letters of known words and the total number of dysphonetic errors. Also, the dyseidetic errors in the known words list contribute to the classification: inversion of similar letters and the total number of dyseidetic errors. In the unknown words list the dysphonetic errors that classified the two groups were: missing double letters, grapheme-phoneme correspondence, omission of letters, grapheme exchange and the total number of dysphonetic errors. No specific dyseidetic errors were selected from the unknown words list but only the total number of dyseidetic errors. Consequently, the total number of dysphonetic and dyseidetic errors in the known and unknown words list was also selected. In contrast to the spelling test, in the reading test there were not many errors that contributed to the classification of the two groups. The most frequent error was confusion of visually similar graphemes both in flash and untimed mode together with the production of non-words in flash mode, dyseidetic errors.

Figure 3 shows a scatterplot with the classification value obtained for each subject in the sample using the final classification equation, divided by groups. For each group a boxplot with the mean and standard deviation of the individual values is presented.

The rAUC for the classification equation was 0.94, with a 0.87 discrimination power at the 10% of False Positive and 0.93 at the 20%. These results are shown in Figure 4.

Left panel in Figure 4 shows the rAUC obtained after the stable classification procedure. The other 3 panels show the distribution density of the total rAUC and the discrimination values at 10% and 20% of False Positive respectively, obtained by the robust procedure for the ROC curve. This consists in 500 random repetitions of the ROC curve calculation with random subsamples, as described in the methodology. As the rAUC, the values at the 50% of the density distributions are selected.

The stable biomarkers identification procedure was also performed with the EEG spectra both at the electrodes and at sources level. However, they did not produce high classification values as it was the case with the DTRS and clinical variables. Therefore, we do not describe them in detail. At the electrodes, the rAUC was 0.73, with a discrimination power of 0.36 at 10% and 0.51 at 20% of False Positive. The variables which participated in the classification equation were Fz and Cz in Delta band; C4 in High Alpha; and F7, Fz, and Pz in Beta band.

At the sources level, the rAUCwas 0.73, with a discrimination power of 0.24 at 10% and 0.53 at 20% of False Positive. The variables which participated in the classification equation were: in Delta band: the Left Middle Frontal lobule, the Left Middle Cingulum and the Right Hippocampus; in Alpha band: the Left Posterior Cingulum; in Beta band: the Right Superior Fronto-Orbital lobule, the Right Inferior Frontal Operculus, the Left Frontal Inferior Triangularis, the Right Middle Cingulum, the Left Posterior Cingulum, the Right Middle Occipital, the Right Heschl, and the Right Superior Temporal Pole.

## 4. Discussion

Statistical comparisons between DD subjects and those with NSRD have led to interesting and important results. The most significant results were found among anamnestic data and those of the DTRS. In particular, subjects with NSRD came to our observation much later than the DD subjects when they were in the last grade of primary school or the first year of secondary school (Table 2). A possible explanation of this delayed bringing to the medical doctors’ attention could lie in the fact that these subjects do not make reading and spelling errors so numerous as to suspect a problem of reading and writing. These children seem overlooked by the teachers because their reading and writing performances differ quantitatively but not qualitatively from those of normal readers and therefore they are referred to medical doctors when they are in the last grade of primary school or much worse when they are at the first grade of secondary school. In fact, these subjects have a spelling test percentage of correctly written words greater than 80. Unlike the reading difficulties present in the dyslexic readers which are often discriminative and clearly contrasting with their abilities and performances, the NSRD have a global performance usually below the norm that is associated with emotional disorders and poor motivation. In fact, the NSRD subjects had a significantly greater number of comorbidities compared to the DD subjects. The most frequent comorbidity after ADHD was the emotional disorder of anxious type. The most common type of comorbidity in the two groups, although the difference was not significant, was ADHD of inattentive type: 47% in the DD subjects and 50% in the NSRD subjects. Although ADHD and SLD are commonly considered to be a separate disorder (DSM-V American Psychiatric Association 2013) many researchers working in this field have observed that dyslexia and ADHD are closely associated. If this association is a true comorbidity or it is, instead, another neurobiological condition it is still subject of discussion and goes beyond the scope of this work.

Another data worthy of attention also comes from the analysis of reading quotients. The subjects with NSRD had a significant lower RQ, RQM and RQC than the subjects with DD. The lower reading quotients in these subjects are explained by the fact that they had a lower reading level and a higher age than the DD subjects.

This observation is also reflected in the mode of reading words in flash presentation. The NSRD subjects had a number of words read significantly more hesitantly than the DD subjects (Table 4). This modality is not typical of dyslexic subjects and in particular of DD subjects. In fact, these subjects prefer to read a word globally as instantaneous visual gestalt, rather than analytically. Their reading, because of their limited sight vocabulary is precipitous; they may guess a word from minimal clues for example from the first or last letter and the length of the word. Again, very often they do not realize the mistakes because they believe that the word they read is not part of their vocabulary. NSRD subjects, on the other hand, seem to be more aware of their difficulties and they do not rush to read a word in flash mode. Other and more numerous significant differences were observed in the second part of the DTRS, the spelling test. DD subjects made significantly more errors than NSRD subjects in both unknown and known word lists. In the list of known words, the most frequent error was the insertion/omission of letters and syllables. This error was also present even more frequently in the list of unknown words (typical error of sequencing of the DD subjects). Other errors of dysphonetic type have been found in the list of unknown words. Lack of double letters, lack of accents, correspondence grapheme/phoneme errors and similar grapheme exchange were significantly more numerous in DD subjects than NSRD. Reading times in both flash and untimed mode when analyzed individually were not significantly different between the two groups. However, when they were analyzed in relation to the complexity index, i.e., number of syllables and number of orthographic difficulties in the word, the DD subjects read much more slowly plurisyllabic words, 4 or 5 syllables, that contained 3 or 4 orthographic difficulties than the NSRD subjects. The fact that with plurisyllabic words, 6 or 7 syllables, these differences were no longer present can be explained by the reduced number of words read in the highest-grade list.

This data allows us to make general methodological and clinical remarks. The lack of significant differences between groups may depend on several reasons: an inaccurate criteria selection of clinical subjects to be examined, the test is not sufficiently demanding for the subjects to bring out differences or more often, the selected test is not appropriate or is not adequately developed to answer either the clinical questions or the model under investigation.

Since the beginning of its publication, the Boder test has received some criticisms, not mainly because of its theoretical construct and clinical acumen, but for the lack of robust normative and validation studies [71,72]. This led to conflicting results and to questioning the existence of the patterns described by Boder. To our opinion, a limitation of the Boder test is that it was developed in the 1960s, when electronic devices were not yet available and therefore it was a paper and pencil test. The flash exposure was obtained by sliding a sheet of paper on a list of words and the classification of instantaneous gestalt reading within a second was entrusted to the experience of the examiner. Therefore, the presentation of words in flash mode was not standardized. Since visual and auditory processing during reading work in parallel, the use of both channels cannot be excluded when an unknown word is presented on paper and consequently it makes difficult to select “unknown” words for the spelling test. This subjectivity may explain the failure to distinguish the reading-spelling patterns and the contradictory results, and it may have introduced differences in the identification of subtypes. The Italian version of the Boder test that we developed in 2001 [3], which is the one used in this paper, is profoundly different from its original version: it is based on normative data built ad hoc, it is a self-paced test, it has been completely automated both in the exposure of words [73] (the words were presented in flash mode only for 250 ms and in untimed mode for 10 s.) and it differs in the recording of reading times that are missing in the original version, being a paper test. The reading and writing errors have been quantitatively and qualitatively analyzed. There is some criticism on the original version of the Boder test about lack of uniformity of its spelling test [71], on which the reading and spelling patterns are primarily based. In the Italian version, being the Italian a transparent language, the selection of words has always followed the same criteria: (a) the known words are chosen at the reading level of the subject and the unknown words are chosen at a higher grade, and (b) they are selected from the words not read or read with great difficulty. In this way we were pretty sure that the words selected for the unknown list were not in the sight vocabulary of the child, adding, therefore, consistency throughout the test.

The subtypes identified in the Boder test have been replicated by many other authors using an indirect approach [74,75,76,77,78,79,80,81]. For a comprehensive revision of this topic, see [5].

The Boder model of dyslexia [1] predicts that in the dyslexic child the normal reading process is dissociated. The normal automatic interplay of gestalt and analytic-synthetic processes is disrupted. The dyslexic child reads and spells differently from the normal reader both qualitatively and quantitatively. In our specific case, to adapt the DTRS to the characteristics of the Italian language and to respect the Boder model we needed to modify the original paper and pencil test: computerization and self-paced reading. The computerization allowed us to present words in flash mode for 250 ms. that in a previous study [73], we proved to be sufficient for a gestalt reading and to record the reading times. With self-paced action we monitor the actual intention of the child to read when pressing a button. Chiarenza et al. [82] demonstrated important electrophysiological differences between a self-paced and an externally paced reading. Even less brain activation was present during passive vision of letters.

Further evidence of the differences between these two groups of subjects comes from the QEEG. As reported by numerous studies already mentioned in the introduction, the main differences consisted of a significant power spectra excess of the DD group in the delta band in the left prefrontal, middle frontal, central, parietal, right parietal and theta power spectra excess in the prefrontal areas bilaterally, and central midline. Additionally, in the same occipital and parietal lobes except the precuneus bilaterally and the right angular gyrus, besides a theta excess, there is also a reduction of beta activity. Excess of slow waves (Delta and Theta) have been related to lower arousal, whereas excess of small beta activity indicates an alert but relaxed state [83].

While excess of delta in children with learning disabilities has been widely reported also recently [84], theta excess has not received unanimous consent. The excess of theta activity in the EEG at rest has been consistently reported in LD-NOS children [16,27,33,85,86,87,88]. Compared with children with good academic achievements, LD-NOS children had evident excess of theta activity (from 3.52 to 7.02 Hz) [86]. However, some authors who have studied this entity have not reported excess in theta activity [11,89,90] although it may be due to the composition of their samples and to the frequency of different types of pathological patterns in the extensive group of LD-NOS. Neurometric QEEG abnormalities have been shown to be directly related to academic performance carefully documented in both reading and writing. Increased delta and/or theta power and decreased alpha power were associated with a poor educational evaluation; increased theta and/or decreased alpha were associated with mildly abnormal evaluations; and increased alpha and decreased theta were associated with good evaluations. Theta excess with alpha deficit was described as reflecting maturational lag, whereas delta excess indicated cerebral dysfunction [33].

The observation of differences of EEG source spectra between the two groups allows us to add further considerations. The first observation concerns the excess of delta and theta activity that is found in the dysphonetic subjects in a standard EEG recorded at rest, with eyes closed. This fact reinforces the hypothesis that dyslexia is not only a functional disorder, but the result of a structural disorder as reported by the studies of Galaburda [91,92,93,94,95,96] that found in the brain of 5 severe dyslexics adults the right temporal planum wider than the left in 100% of cases. In addition, a high frequency of microdysgenesis was also observed, particularly in the left frontal and temporal opercula [91,97]. This report was subsequently confirmed by Shaywitz et al. [98] who performed a series of language-based activation tasks with progressively increasing phonologic demands using fMRI in dyslexic adults. There was underactivation of the left posterior perysilvian and occipital regions (Wernicke’s area, the angular gyrus and striate cortex) and overactivation to even simple phonologic task in both left anterior (inferior frontal gyrus) and right posterior perysilvian regions (see also for a review on this topic [99]). Together with the aforementioned brain areas, Wood and Flowers [100], with a factor analytic validation across 100 cases (50 normal and 50 dyslexic), using positron emission tomography have identified 43 candidate regions involved in reading and writing. The EEG sources identified with our method (Figure 2) confirm what was already found with other neuroimaging methods. To those cerebral areas already described the superior, middle and medial frontal areas show an excess of theta. A certain dysregulation of the motor areas in dyslexic subjects has long been known [101,102]. Dyslexic patients have difficulty processing both rapid and visual stimuli as well as in generating rapid bimanual motor output. Chiarenza et al. [103,104,105] and Chiarenza [106] recorded the brain electrical activity, called "movement-related brain potentials", during a skilled motor task that to be performed adequately, required bimanual coordination, bimanual ballistic movements, adaptive programming and learning a proper timing. The dyslexic children presented a deficit of programming movements, a deficit of visual and kinesthetic sensory processes, and a reduced capacity to evaluate their performance and correct their errors. Chiarenza [46] advanced the hypothesis that dyslexia is not only a phonological or gestalt deficit, but also a praxic disorder in which praxic abilities, such as motor programming, sequential and sensorial motor integration and evaluation processes, are required and somehow defective in dyslexia.

In addition, Chiarenza [106] has observed that dyslexic children showed a latency delay of movement-related potentials significantly different among the various cerebral areas during the same skilled motor task. Therefore, Chiarenza [106] has hypothesized that dyslexia could be the result of a timing defect that causes an integration defect and dysfunction of numerous processes hierarchically organized that occur at different levels and times. Also, Llinas [107] hypothesizes that at the base of the pathophysiology of dyslexia exists a more basic deficit of timing. This dyschronia results from a cellular dysfunction that modifies “the normal properties of neuronal circuits responsible for temporal aspects of cognition” so that the nervous system can function in a relatively normal fashion only within a temporal window. Llinas postulates that for some reason these neurons are unable to generate sharp enough ensemble oscillations at higher frequencies and reset such rhythmicity following close-interval sensory stimulation. Moreover, the dyschronia does not apply to a particular cerebral locus, but it is much wider and involves the entire brain network involved in learning and reading. Likewise, the same timing concept can be applied to the various EEG frequency bands. Different frequencies favor different types of connections and different levels of computation. In general, slow oscillations can involve many neurons in large brain areas, whereas the short time windows of fast oscillators facilitate local integration [108]. Reading, as Boder claims, requires the perfect dynamic interplay of intact visual-gestalt and analytic-auditory function and integration of both peripheral and central processes. Therefore, the excess of delta and theta found in our dysphonetic subjects could reflect a temporal dysregulation already observed with other methodologies at various brain levels and in different cerebral areas.

This dynamic interplay of different frequencies in different brain areas confirms the idea of how the process of reading and writing occurs through a complex network in different brain areas and therefore it seems plausible that an alteration at one point in the network is inevitably reflected in other areas of the brain. A possible explanation of this temporal dysregulation could be explained as a compensatory mechanism or, alternatively, to the fact that in the dysphonetic subjects the normal automatic interplay of gestalt and analytic-synthetic processes is interrupted. The DD subjects tend to persist in the gestalt approach preferring to guess at unfamiliar words rather than employ their word-analysis skills.

Bosch-Bayard et al. [34] have given strong evidence of the validity and robustness of the biomarkers selection procedure based on the GLMNet method. GLMNet itself has also been successfully applied in different clinical situations [109,110,111]. In particular, the variables that had the best discriminating power were those of the DTRS and the anamnestic data, with a value of 0.94 for the stable area under the ROC curve. This very high discrimination also had a high specificity at a value of 10% of False Positives (see Figure 4). Therefore, the DTRS appears as a very useful tool for the differentiation of the DD regarding the NSRD. Among the anamnestic data, the FSIQ, VIQ and the speech therapy had a discriminating power in addition to the age and class at the time of the administration of DTRS. The FSIQ, VIQ and speech therapy had a frequency rate of 78.5, 86.5 and 96.3, respectively (see Table 6). The verbal subscales of WISCIII evaluates the quality and quantity of vocabulary and information. Due to their poor vocabulary the DD subjects scored lower than the NSRD subjects. They also had a history of speech therapy because of their obvious difficulties.

Table 6 also shows that errors in reading and spelling contributed significantly to the classification of the two groups. In the reading test, the biggest differences between the two groups were the confusion of visually similar graphemes both in flash and untimed mode and the production of non-words in flash mode.

The list of known words of the spelling test, constructed with words that are part of the subject’s sight vocabulary, is intended to evaluate the child’s ability to ‘revisualize’ the words in his sight vocabulary. The lists of unknown words, made with words that the subjects have not been able to read or have read with great difficulty, evaluate the child’s ability to spell words not in his sight vocabulary. From the list of known words, mismatches of grapheme-phoneme, omission of letters, and reversal of similar letter were selected as biomarkers. From the list of unknown words, errors of accentuation, lack of double, omission of letters, exchange of similar graphemes were also selected. The DD subjects had a greater number of errors compared to NSRD subjects.

Previous studies have found that dysphonetic and dyseidetic children have almost identical profiles. Clustering techniques using neuropsychological and psychoeducational measures have yielded fairly homogeneous clusters corresponding to the proposed dysphonetic and dyseidetic subtypes [72,112,113,114]. Our findings indicate that there are significant differences between DD and NSRD groups at the neuropsychological level. Additionally, our classification procedure mostly selected dysphonetic errors and only two dyseidetic errors, which are present only in the reading test: confusion of visually similar graphemes both during flash and untimed mode. The minus sign of the coefficients in Table 6 indicates that subjects with NSRD had more errors of this type than DDs.

On the other hand, the EEG did not exhibit a high classification power of the two groups. The EEG spectra were analyzed both at the leads as well as at the sources in the whole frequency range of the HSR model. However, neither of the two equations obtained high discrimination power with the stable ROC technique (0.73 for the EEG spectra at the scalp and 0.73 for the spectra at the sources, see Section 3.3). These results point out that even when electrical activity between the two groups show differences, they do not clearly discriminate them with high sensitivity and specificity.

## 5. Conclusions

In this work we have described the clinical and neurophysiological characteristics that distinguish DD subjects from NSRD. The DTRS proved to be the one with the most discriminating power. The number and type of writing errors are those that better differentiate the two groups. The constant presence of dysphonetic and dyseidetic errors in subjects at the end of the first cycle of primary school must promptly draw the attention of teachers who must request a thorough clinical evaluation for the obvious therapeutic and prognostic implications.

## Figures and Tables

**Figure 1 brainsci-08-00172-f001:**
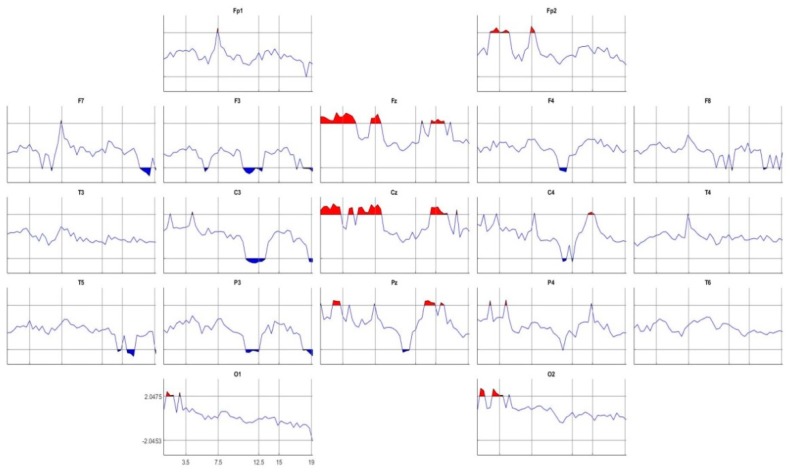
Results of the *t*-test between DD and NSRD groups for the Z-EEG spectra at all leads and frequencies. Main significant differences are an excess of activity of the DD group in the middle line in almost all frequency bands and a reduction of fast activity in the left fronto-parietal and temporal leads.

**Figure 2 brainsci-08-00172-f002:**
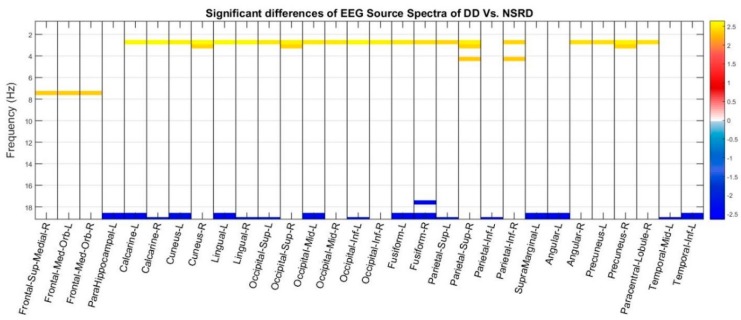
Significant differences of the *t* test at the sources spectra of DD vs NSRD. Red and yellow values indicate an excess of DD compared to NSRD; values in the blue scale indicates excess of NSRD compared to DD. Threshold corrected by multiple comparisons. Differences are concentrated in very narrow bands of frequencies. In general, DD have an excess of slow activity (Delta and slightly Theta) and a defect of fast activity (Beta band).

**Figure 3 brainsci-08-00172-f003:**
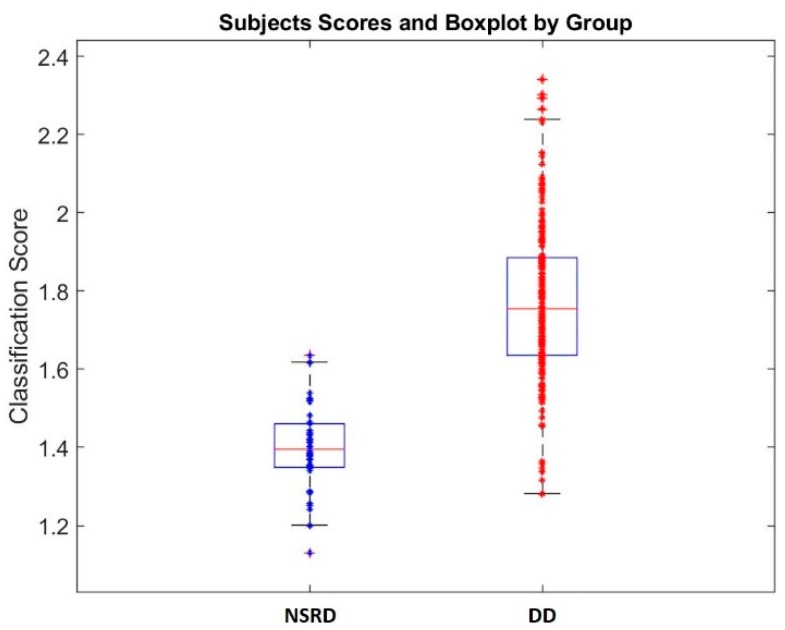
Scatterplot showing the individual scores obtained by the classification equation of the DTRS variables, for the subjects of the two groups NSRD (unspecific reading delay) and DD (dysphonetic dyslexics). For each group a boxplot with the mean and standard deviation of the individual values is shown.

**Figure 4 brainsci-08-00172-f004:**
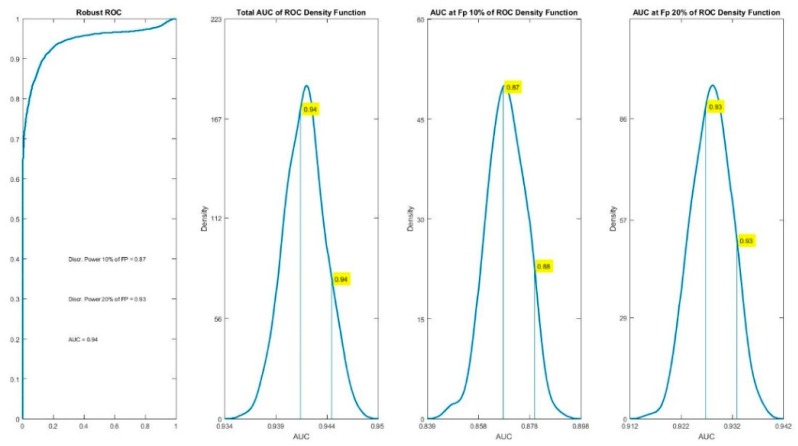
rAUC for the classification equation showing a value of 0.94, which indicates almost perfect sensitivity and specificity. The three panels at the right show the density functions of the total rAUC as well as the discrimination power at the 10% and 20% of False Positives, obtained by the 500 random realizations. The discrimination power at the 50% of the density function was of 0.87 for the 10% of False Positives and 0.93 at the 20% of False Positives.

**Table 1 brainsci-08-00172-t001:** *t*-test results of the anamnestic data of the two groups: mean, SD, *t* value and *P* value. DD. = Dysphonetic dyslexia, NSRD = Non-Specific Reading Delay, N.S. = number of subjects, SD = standard deviation, FSIQ = full scale intelligent quotient, VIQ = verbal intelligent quotient, PIQ = performance intelligent quotient, S.L.D. = specific language disorder, Dysorth. = Dysorthography, Dysgr. = dysgraphia, Dyscal. = dyscalculia, Com. = comorbidity, T.O.C. = type of comorbidity, S.T. = speech therapy, T.O.S.T. = type of speech therapy, S.T.D. = speech therapy duration, F.Q. = father qualification, M.Q. = mother qualification.

	FSIQ	VIQ	PIQ	S.L.D.	Dysorth.	Dysgr.	Dyscal.	Com.	T.O.C.	S.T.	T.O.S.T.	S.T.D.	F.Q.	M.Q.
DD (N.S. = 169)—Means	101.4	99.6	103.4	0.57	0.6	0.21	0.31	0.25	6.48	0.19	0.08	3.36	2.27	2.34
NSRD (N.S. = 36)—Means	105.5	103.9	106.6	0.58	0.49	0.22	0.27	0.44	6.5	0.04	0	0.31	2.38	2.22
DD—SD	10.9	11,7	12,1	0.61	0.49	0.41	0.46	0.43	1.66	0.39	0.5	9.04	2.39	2.36
NSRD—SD	9.2	12.1	10.0	0.54	0.51	0.42	0.45	0.5	0.61	0.21	0	1.47	2.26	2
DD Vs. NSRD—*P* value	0.02	0.03	0,1	0.93	0.18	0.88	0.59	0.008	0.95	0.01	0.29	0.025	0.79	0.75
DD Vs. NSRD—*t* value	−2.3	−2.19	−1.64	−0.08	1.32	−0.14	0.53	−2.7	−0.06	2.4	1.04	2.25	−0.26	0.31

**Table 2 brainsci-08-00172-t002:** Means, SD, *t*-test results of the reading test of DTRS in the two groups. DD = dysphonetic dyslexia, NSRD = Unspecific reading delay, SD=standard deviation, N.L.R. = Number of lists read, RL = reading level, RA = reading age, RQ = reading quotient related to chronological age, RQM = reading quotient related to mental age, RQC = reading quotient related to mental and chronological age.

	School	Age at DTRS	N.L.R.	RL	RA	RQ	RQM	RQC
DD (N.S. = 169)—Means	1.26	10.02	6.72	2.38	8.07	80.58	79.99	80.08
NSRD (N.S. = 36)—Means	1.47	10.84	6.87	2.27	8.12	75.23	71.64	73.26
DD—SD	0.46	1.92	2.49	1.38	1.79	9.63	9.82	8.64
NSRD—SD	0.59	2.17	2.3	1.5	1.82	10.04	9.98	9.52
DD Vs. NSRD—*P* value	0.011	0.013	0.72	0.66	0.88	0.001	7.8 × 10^−7^	6.1 × 10^−6^
DD Vs. NSRD—*t* value	−2.56	−2.51	−0.36	0.43	−0.15	3.31	5.08	4.64

**Table 3 brainsci-08-00172-t003:** *t*-test results of the spelling test of the DTRS of the two groups: mean, SD, *t* value and *P* value. N.S. = number of subjects, %KW = % known words, %UW = % unknown words, N.D.E.KW = number of dysphonetic errors of known words, D.E.IOL.UW = dysphonetic errors-insertion/omission of letters of known words, N.DY.E.KW = number of dyseidetic errors of known words, N.D.E.UW = number of dysphonetic errors of unknown words, D.E.MDL.UW = dysphonetic errors of missing double letters of unknown words, D.E.A.UW = dysphonetic errors of accent of unknown words, D.E.GP.UW = dysphonetic errors correspondence grapheme-phoneme of unknown words, D.E.IOL.UW = dysphonetic errors-insertion/omission of letters of unknown words, D.E.MG.UW = dysphonetic errors-mismatch of grapheme of unknown words.

	%KW	%UW	N.D.E.KW	D.E.IOL.KW	N.DY.E.KW	N.D.E.UW	D.E.MDL.UW	D.E.A.UW	D.E.GP.UW	D.E.IOL.UW	D.E.MGUW
DD (N.S. = 169)—Means	71.81	47.65	1.9	1.06	1.2	5.1	1.35	0.28	1.51	1.21	0.56
NSRD (N.S. = 36)—Means	89.44	83.61	0.6	0.29	0.4	1.7	0.3	0.11	0.5	0.55	0.2
DD—SD	18.68	15.5	1.6	1.17	1.4	2.4	1.46	0.53	1.54	1.23	0.13
NSRD—SD	7.43	5.84	0.7	0.51	0.5	1	0.51	0.31	0.64	0.68	0.34
DD Vs. NSRD *P* value	2.8 × 10^−10^	1.4 × 10^−33^	3.2 × 10^−7^	2.4 × 10^−5^	1.0 × 10^−10^	1.5 × 10^−17^	4.8 × 10^−6^	0.012	1.2 × 10^−5^	0.001	0.008
DD Vs. NSRD *t* value	−6.61	−14.38	5.27	4.31	3.55	9.29	4.69	2.53	4.47	3.3	2.68

**Table 4 brainsci-08-00172-t004:** *t*-test results of the reading test of DTRS of the two groups: mean, SD, *t*-value and *p*-value. SD = standard deviation, FCOR = number of words correctly read in flash mode, FINCORCOR = number of words incorrectly read in flash mode, FHESIT = number of words hesitating read in flash mode, FSILL = number of words syllabicating read in flash mode, UCOR = number of words correctly read in untimed mode, UINCOR = number of words incorrectly read in untimed mode, UGDIFF = number of words read with great difficulty in untimed mode, NR = number of words not read, RTFCOR = averaged reading time of words correctly read in flash mode, RTFINCORCOR = average reading time of words incorrectly read in flash mode, RTFHESIT = average reading time of words hesitating read in flash mode, RTFSILL = average reading time of words syllabicating read in flash mode, RTUCOR = average reading time of words correctly read in untimed mode, RTUINCORCOR = average reading time of words incorrectly read in untimed mode, RTUGDIFF = average reading time of words read whit great difficulty in untimed mode.

	FCOR	FINCOR	FHESIT	FSILL	UCOR	UINCOR	UGDIFF	NR	RTFCOR	RTFINCOR	RTFHESIT	RTFSILL	RTUCOR	RTUINCOR	RTUGDIFF
DD (N.S. = 169)—Means	15.1	2	1.8	0.2	2.7	0.2	0.3	0	1	2	2	1.26	1.6	1.7	3.3
NSRD (N.S. = 36)—Means	14.6	2	3	0.1	2.1	0	0.1	0	1	1,9	1.9	1.84	1.6	2	3.2
DD—SD	4.5	2.2	2.6	0.7	4	0.5	0.8	0.2	0.5	1.1	1	0.77	0.7	1.9	1.9
NSRD—SD	4.2	2.4	4	0.5	2.6	0.2	0.3	0	0.5	1.1	1	0.25	0.9	1.9	1.9
DD Vs. NSRD—*P* value	4.49	0.65	0.01	0.3	0.27	0.12	0.14	0.50	0.33	0.90	0.64	0.22	0.90	0.50	0.70
DD Vs. NSRD—*t* value	4.49	0.46	−2.55	1.05	1.11	1.55	1.49	0.68	0.97	0.12	0.47	−1.25	0.16	−0.68	0.38

**Table 5 brainsci-08-00172-t005:** Significant differences of DD vs. NSRD in reading time according to the reading mode (by columns) compared to the words difficulty (by rows). The rows name indicates: the first digit, the number of syllables of the word and the second digit, the number of grammatical difficulties. By columns: 11: Flash-correct; 15: Flash-incorrect; 21: sustained correct; 22: sustained incorrect; 30: non-read. The non-zero cells contain the t-test value above and the *P* value below. The reading time of the DD children was in general higher than NSRD, except for the combination of 21–32.

	10	11	12	20	21	22	23	30	31	32	33	40	41	42	43	44	50	51	52	53	60	61	62	63	70	71
11				2.92 0.004																						
15																										
21				2.21 0.028						−2.38 0.019						2.64 0.025				2.41 0.024						
22																				9.35 0.003						
30																										

**Table 6 brainsci-08-00172-t006:** Biomarkers obtained from the classification procedure with the Clinical and Anamnestic variables (first column). The second column indicates the percent of the times the variables were selected as biomarkers during the random classification procedure. The third column is the coefficient that accompany each variable in the final classification equation. The stable area under the ROC for this equation is 0.94.

Variables	Percent	Coeff.
Anamnestic and clinical data		
Class	100	0.031413
Comorbidity	100	0.029233
Speech therapy	96.36	0.053496
Verbal Intelligent Quotient	86.57	0.025448
Age at DTRS	83.56	0.029181
School at DTRS	80.85	−0.00266
Full Scale Intelligent Quotient	78.57	0.005488
DTRS: Writing Test: Dysphonetic errors in the known words list		
Grapheme-phoneme correspondence	77.59	0.049424
Omission of letters	76.67	0.005422
Number of dysphonetic errors	76	0.016698
DTRS: Writing Test: Dyseidetic errors in the known words list		
Errors of reversal of similar letters	72.13	−0.001226
Number of dyseidetic errors	70.49	0.010814
DTRS: Writing Test: Dysphonetic errors in the unknown words list		
Missing double letters	67.86	0.000454
Accent errors	67.35	0.004261
Grapheme-phoneme correspondence	62.5	0.03555
Omission of letters	62	0.054781
Grapheme mismatch	60.38	−0.034673
Number of dysphonetic errors	59.18	0.018046
DTRS: Writing Test: Dyseidetic errors in the unknown words list		
Total dyseidetic errors	55.17	−0.009407
Total errors in the known and unknown words lists combined		
Total dysphonetic errors	54.39	0.021067
Total dyseidetic errors	52.54	−0.005514
DTRS: Reading Test: Dyseidetic errors in flash mode		
Confusion of visually similar graphemes	51.52	−0.019377
Dyseidetic non-words	50.75	0.016049
DTRS: Reading Test: Dyseidetic errors in untimed mode		
Confusion of visually similar graphemes	50.75	−0.03763

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
