# Peer review of "Clinical and Electrophysiological Differences between Subjects with Dysphonetic Dyslexia and Non-Specific Reading Delay"

_brainsci, 2018, doi:10.3390/brainsci8090172_

Round 1
Reviewer 1 Report
This study is very interesting. It potentially demonstrates that there are objective differences between dyslexic and non specific reading delay (NSRD). This is highly relevant to the current dispute about whether there are any real differences between them, although this issue is hardly discussed. But it has several other weaknesses.
First, there is a large literature disputing the Boder classifications which is hardly discussed, and then hardly used as the analysis is almost exclusively on the 'dysphonetic' type.
Second, the analyses of the DTRS and QEEG, separately or combined are very difficult to understand - and not helped by the seemingly random use of NSRD and UNRD which I take to be identical.
Third the seemingly arbitrary delta and theta frequency differences should be put in the context of what is know of their physiological significance and how these may relate to the brain areas know to be involved in the reading process.
Fourth, and perhaps most importantly, the English needs thorough revision.
Author Response
We want to thanks to the reviewer for the positive opinions and the very helpful comments. Here we post our answers. In yellow we mark our comments to the reviewer, and in cyan we mark the specific text that have been added to the paper, in response to the reviewers’ criticism.
Comments and Suggestions for Authors
This study is very interesting. It potentially demonstrates that there are objective differences between dyslexic and non specific reading delay (NSRD). This is highly relevant to the current dispute about whether there are any real differences between them, although this issue is hardly discussed. But it has several other weaknesses.
ANSWER:
We have emphasized the differences between the DD and NSRD in different parts in the text. Specifically, we have added the following paragraph:
Previous studies have found that dysphonetic and dyseidetic children have almost identical profiles. Clustering techniques using neuropsychological and psychoeducational measures have yielded fairly homogeneous clusters corresponding to the proposed dysphonetic and dyseidetic subtypes [99] McIntosh & Gridley 1993, Swanson et al. 1990, Watson and Willows 1995. Our findings of a discrimination equation which such a high percent of accuracy indicates that there are significant differences between DD and NSRD groups at the neuropsychological level. Additionally, our classification procedure of spelling errors of the list of unknown words mostly selected dysphonetic type of error and only two variables of dyseidetic type errors and they are present only in the reading test as a confusion of visually similar graphemes both during flash and untimed mode. The minus sign of the coefficients in Table 6 indicates that subjects with UNRD had more errors of this type than DDs.
First, there is a large literature disputing the Boder classifications which is hardly discussed, and then hardly used as the analysis is almost exclusively on the 'dysphonetic' type.
ANSWER:
We thank the referee 1 for the note on the Boder test that allows us to make further considerations.
We have added the controversy about Boder classification, cited the references and discussed the issue including 13 new references, as follows:
Since the beginning of its publication the Boder test has received some criticisms mainly because of its theoretical construct and clinical acumen, but for the lack of robust normative and validation studies [98,99]. This led to conflicting results and to questioning the existence of the patterns described by Boder. To our opinion, a limitation of the Boder test is that it was developed in the 1960s when electronic devices were not yet available and therefore it was a paper and pencil test. The flash exposure was obtained by sliding a sheet of paper on a list of words and the classification of instantaneous gestalt reading within a second was entrusted to the experience of the examiner. Therefore, the presentation of words in flash mode was not standardized. Since visual and auditory processing during reading work in parallel, the use of both channels cannot be excluded when an unknown word is presented on paper and consequently it makes difficult to select “unknown” words for the spelling test. This subjectivity may explain the failure to distinguish the reading-spelling patterns and the contradictory results and may have introduced differences in the identification of subtypes. The Italian version of the Boder test that we developed in 2001 [20], which is the one used in this paper, is profoundly different from its original version: it is based on normative data built ad hoc, it is a self-paced test, it has been completely automated both in the exposure of words [103] (the words were presented in flash mode only for 250 ms and in untimed mode for 10 s.) and in the recording of reading times that are missing in the original version, being a paper test. The reading and writing errors have been quantitatively and qualitatively analyzed. In relation to some criticism on the original version of the Boder test and in particular about the non-uniformity of the spelling test [98], on which are primarily based the reading and spelling patterns, being the Italian a transparent language, the selection of words, in the Italian version, has always followed the same criteria, a) the known and unknown words were chosen at the reading level of the subject or a grade higher, b) they were selected from the words not read or read with great difficulty. In this way we were pretty sure that the words selected for the unknown list were not in the sight vocabulary of the child, adding, therefore, consistency throughout the test.
The subtypes identified in the Boder test have been replicated by many other authors using an “indirect” approach [104-111]. For a comprehensive revision of this topic, see [19].
Methodologically, we describe the complete Boder classification algorithm and briefly mention all the groups described by Boder. However, in this paper we concentrate only in two of the subgroups: the dysphonetics (DD), which is the more frequent subtype and the Non-specific Reading Delay (NSRD), which is the less diagnosed and we try to find objectives differences between them. The analysis of the dyseidetics and the mixed subgroups will be the focus of another work. We have also mentioned this in the text. We also added more detail about the Italian version of the Boder test, as follows:
Two independent researchers conducted both a qualitative and quantitative analysis of the errors of the reading and spelling test of the subjects of the entire sample according to the classification proposed by Boder (1973) with the addition of some errors typical of the Italian language as accents and double letters.
At the end of the reading test the computer automatically provides the reading level (‘RL’), the reading age (‘RA’), and the reading quotient (‘RQ’): the ratio between ‘RA’ and chronological age (CA) (RQ = (RA/CA) x 100). If the child’s overall mental ability is substantially above or below average, this quotient is corrected for mental age (MA) by use of the following formula, RQM = (RA/MA) x 100. The RQC is the reading quotient corrected for mental and chronological age according to the following formula, RQC = (2RA/(MA + CA) x 100.
and discussed the importance of these modifications as follows:
Since the beginning of its publication the Boder test has received some criticisms mainly because of its theoretical construct and clinical acumen, but for the lack of robust normative and validation studies [98,99]. This led to conflicting results and to questioning the existence of the patterns described by Boder. To our opinion, a limitation of the Boder test is that it was developed in the 1960s when electronic devices were not yet available and therefore it was a paper and pencil test. The flash exposure was obtained by sliding a sheet of paper on a list of words and the classification of instantaneous gestalt reading within a second was entrusted to the experience of the examiner. Therefore, the presentation of words in flash mode was not standardized. Since visual and auditory processing during reading work in parallel, the use of both channels cannot be excluded when an unknown word is presented on paper and consequently it makes difficult to select “unknown” words for the spelling test. This subjectivity may explain the failure to distinguish the reading-spelling patterns and the contradictory results and may have introduced differences in the identification of subtypes. The Italian version of the Boder test that we developed in 2001 [20], which is the one used in this paper, is profoundly different from its original version: it is based on normative data built ad hoc, it is a self-paced test, it has been completely automated both in the exposure of words [103] (the words were presented in flash mode only for 250 ms and in untimed mode for 10 s.) and in the recording of reading times that are missing in the original version, being a paper test. The reading and writing errors have been quantitatively and qualitatively analyzed. In relation to some criticism on the original version of the Boder test and in particular about the non-uniformity of the spelling test [98], on which are primarily based the reading and spelling patterns, being the Italian a transparent language, the selection of words, in the Italian version, has always followed the same criteria, a) the known and unknown words were chosen at the reading level of the subject or a grade higher, b) they were selected from the words not read or read with great difficulty. In this way we were pretty sure that the words selected for the unknown list were not in the sight vocabulary of the child, adding, therefore, consistency throughout the test.
Second, the analyses of the DTRS and QEEG, separately or combined are very difficult to understand - and not helped by the seemingly random use of NSRD and UNRD which I take to be identical.
ANSWER:
Sorry for the inconsistency. UNRD has been substituted by NSRD everywhere in the text.
Additionally, the reviewer is right, the biomarkers analysis of QEEG and DTRS variables is very confusing and it was our mistake. Our primary purpose was to find the best classification equation with the higher discrimination power. We planned to test both: the clinical data and the QEEG data separated and combined in the same equation to try to get the best classification power. And we wrote that in our first draft. However, when we analyzed the clinical data itself, the classification power was 0.94 percent. This is almost perfect, just one subject was misclassified. Therefore, it didn’t make any sense for us to try the combination of the two datasets. Instead, we analyzed the two datasets separately and compared the power of each of them. But we forgot to remove this sentence about the combination from the draft. We apologize for that and we have removed the sentence from the draft and other mentions to that combination in other parts.
Third the seemingly arbitrary delta and theta frequency differences should be put in the context of what is know of their physiological significance and how these may relate to the brain areas know to be involved in the reading process.
ANSWER:
We have modified the discussion to better emphasize the possible meaning of the Delta, Theta and Beta differences between the two groups. In particular, we have added two paragraphs (together with the whole discussion) to reflect this idea:
Further evidence of the differences between these two groups of subjects comes from the QEEG. As reported by numerous studies already mentioned in the introduction the main differences consisted of a significant power spectra excess of the DD group in the delta band in the left prefrontal, middle frontal, central, parietal, right parietal and theta power spectra excess in the prefrontal areas bilaterally, and central midline. Additionally, in the same occipital and parietal lobes except the precuneus bilaterally and the right angular gyrus, at the same time, besides a theta excess, there is also a reduction of beta activity. Excess of slow waves (Delta and Theta) have been related to lower arousal, whereas excess of small beta activity indicates an alert but relaxed [115].
And:
In addition, Chiarenza (1990) has observed that dyslexic children showed latency delay of movement cognitive potentials significantly different among the various cerebral areas during the same skilled motor task. Therefore, Chiarenza has hypothesized that dyslexia could be the result of a timing defect that causes an integration defect and dysfunction of numerous processes hierarchically organized that occur at different levels and times. Also, Llinas (1993) hypothesizes that at the base of the pathophysiology of dyslexia exists a more basic deficit of timing. This dyschronia results from a cellular dysfunction that modifies “the normal properties of neuronal circuits responsible for temporal aspects of cognition” so that the nervous system is allowed to function in a relatively normal fashion only within a particular temporal window. Llinas postulates that for some reason these neurons are unable to generate sharp enough ensemble oscillations at higher frequencies and reset such rhythmicity following close-interval sensory stimulation. Moreover, the dyschronia does not apply to a particular cerebral locus, but it is much wider and involves the entire brain network involved in learning and reading. Likewise, the same timing concept can be applied to the various EEG frequency bands. Different frequencies favor different types of connections and different levels of computation. In general, slow oscillations can involve many neurons in large brain areas, whereas the short time windows of fast oscillators facilitate local integration (Buzsáki, 2006). Reading, as Boder claims, requires the perfect dynamic interplay of intact visual-gestalt and analytic-auditory function and integration of both peripheral and central processes. Therefore, the excess of delta and theta found in our dysphonetic subjects could reflect a temporal dysregulation already observed with other methodologies at various brain levels and in different cerebral areas.
Please, read the discussion again to get the whole sense about this topic.
Fourth, and perhaps most importantly, the English needs thorough revision.
ANSWER:
Thank you! We have made a revision of all the text and corrected many grammatical and typing errors we have found. Also, a native English speaker revised and corrected the text.
Reviewer 2 Report
The authors examined the neuropsychological and EEG profiles of dysphonetic dyslexics (DD) vs. individuals with non-specific reading disorders (NSRD). They found cross-group differences in both types of measures. The authors have further tested the ability of neuropsychological and EEG measures to predict participants’ pertinence to one of the two groups (discriminative power) and found strong discriminative power in the former (neuropsychological), but not the latter (EEG).
General appreciation
The problem that is addressed - diagnosing subtypes of reading difficulties - is relevant. Data analyses seem detailed and well-documented. However, I found it difficult to figure out what is the main scientific question, and which answer the authors provided to it (see below). I think the authors should clarify these issues before publication.
Major concern
After highlighting the potential of QEEG for discriminating reading-difficulties subtypes, the authors predict the following: “It is expected that the combined use of clinical indices together with the EEG spectra at the sources can effectively discriminate the two groups (lns 134-136).” My questions are:
- As far as I understood, the DTRS is already used to discriminate DD from NSRD. The authors used this test themselves to create the subsamples. So what is the advantage of adding EEG combined with the DTRS in the future? Is it to improve discrimination? In that case, did the authors show that discrimination is improved by adding the EEG (I didn’t get that impression)?
- Although the prediction was stated as abovementioned, the authors state: “The objective of this research is to report the results of neurophysiological and neuropsychological differences between subjects with dyslexia and with NSRD (lns 136-138)”. The truth seems that the authors did both: they (1) examined the differences between groups, and they (2) tested whether neuropsychological and/or EEG data discriminate the two groups. For (1), they found differences in neuropsychological and EEG data, for (2) they found that neuropsychological variables predict group pertinence, EEG does not, and some EEG combined with neuropsychological variables keeps predicting as well as neuropsychological alone. So, depending on the goal, we have different outcomes. It is possible, however, that I am missing something.
- I think that “reporting the results of group differences” may be a too-low-level formula for stating a goal (all research papers report results, and most of them of group differences; the question is for what).
- If DTRS was used to create the subsamples, is it not expected that the DTRS predicts pertinence to each sample? Or was it that the author’s goal was determining which DTRS components best predict pertinence?
Minor issues
Table information could be condensed (too many tables, too much information per table)
There are some formatting problems, e.g.
Ln 20- a dot is missing; Ln 32 – extra space; Ln 64-comma missing; ln 106 – space missing, etc…
Author Response
We want to thanks to the reviewer for the positive opinions and the very helpful comments. Here we post our answers. In yellow we mark our comments to the reviewer, and in cyan we mark the specific text that have been added to the paper, in response to the reviewers’ criticism.
Comments and Suggestions for Authors
The authors examined the neuropsychological and EEG profiles of dysphonetic dyslexics (DD) vs. individuals with non-specific reading disorders (NSRD). They found cross-group differences in both types of measures. The authors have further tested the ability of neuropsychological and EEG measures to predict participants’ pertinence to one of the two groups (discriminative power) and found strong discriminative power in the former (neuropsychological), but not the latter (EEG).
General appreciation
The problem that is addressed - diagnosing subtypes of reading difficulties - is relevant. Data analyses seem detailed and well-documented. However, I found it difficult to figure out what is the main scientific question, and which answer the authors provided to it (see below). I think the authors should clarify these issues before publication.
ANSWER:
Now we have clarified the main scientific question in the text as follows:
The purpose of this research was to find objectives differences between DD and NSRD groups, both at the neurophysiological and neuropsychological levels. At the same time, to find a classification equation which discriminates the two groups with a high percent of accuracy, which could contribute to the earlier diagnose of the NSRD group, usually very late diagnosed and therefore, untreated until the secondary school level. This may provide clinicians and therapists with alarm indices deriving from the anamnesis and the results of the DTRS that should lead to an earlier diagnosis of reading delay.
Major concern
After highlighting the potential of QEEG for discriminating reading-difficulties subtypes, the authors predict the following: “It is expected that the combined use of clinical indices together with the EEG spectra at the sources can effectively discriminate the two groups (lns 134-136).” My questions are:
ANSWER:
The reviewer is right. Our primary purpose was to find the best classification equation with the higher discrimination power. We planned to test both: the clinical data and the EEG data separated and combined in the same equation to try to get the best classification power. And we wrote that in our first draft. However, when we analyzed the clinical data itself, the classification power was 0.94 percent. This is almost perfect, just one subject was misclassified. Therefore, it didn’t make any sense for us to try the combination of the two datasets. Instead, we analyzed the two datasets separately and compared the power of each of them. But we forgot to remove this purpose from the draft. We apologize for that and we have removed the sentence from the draft and other mentions to that combination in other parts.
- As far as I understood, the DTRS is already used to discriminate DD from NSRD. The authors used this test themselves to create the subsamples. So what is the advantage of adding EEG combined with the DTRS in the future? Is it to improve discrimination? In that case, did the authors show that discrimination is improved by adding the EEG (I didn’t get that impression)?
ANSWER:
We already clarified the confusion with the combination of the EEG and DTRS data. It was simply a mistake forgotten in the draft. With respect to the use of the DTRS both as the instrument for classification and then again as a source of data for the classification algorithm, it is in fact confusing. The Boder classification algorithm does not use all items of the DTRS for the classification. We did not include those items in the biomarkers selection algorithm. What we found was a combination of clinical items, not used for the Boder classification, which almost perfectly separates the two groups. This is what we say that may be used in the future to create a simpler instrument for the discrimination of these two groups and at the same time provide clinicians and therapists with alarm indices deriving from the anamnesis and the results of the DTRS that should lead to an earlier diagnosis of reading delay. We have clarified that in the text, as follows:
Since the DTRS is the instrument used in the Boder algorithm to classify the subjects into the DD and NSRD groups, we only included in the biomarkers selection procedure those DTRS items which were not used in the Boder algorithm (% of known words, % of unknown words, RL, RA, RQ, RQM, RQC). Therefore, in the case of the DTRS data, we found a set of biomarkers completely independent to those used for the creation of the dyslexia subgroups.
- Although the prediction was stated as abovementioned, the authors state: “The objective of this research is to report the results of neurophysiological and neuropsychological differences between subjects with dyslexia and with NSRD (lns 136-138)”. The truth seems that the authors did both: they (1) examined the differences between groups, and they (2) tested whether neuropsychological and/or EEG data discriminate the two groups. For (1), they found differences in neuropsychological and EEG data, for (2) they found that neuropsychological variables predict group pertinence, EEG does not, and some EEG combined with neuropsychological variables keeps predicting as well as neuropsychological alone. So, depending on the goal, we have different outcomes. It is possible, however, that I am missing something.
ANSWER:
The reviewer is again right. The summary made by the reviewer in (1) and (2) is correct. Our discussion was permeated by our a priori expectations. We did not expect to get such excellent results for the DTRS data and we kept talking about the combination in the text, which was unnecessary. That was a mistake and we have corrected the text accordingly.
- I think that “reporting the results of group differences” may be a too-low-level formula for stating a goal (all research papers report results, and most of them of group differences; the question is for what).
ANSWER:
Thank you. We have changed this phrase in the text. We have considered the valuable opinions of the two reviewers to rephrase our statement. Now it is written as follows:
The purpose of this research was to find objectives differences between DD and NSRD groups, both at the neurophysiological and neuropsychological levels. At the same time, to find a classification equation which discriminates the two groups with a high percent of accuracy, which could contribute to the earlier diagnose of the NSRD group, usually very late diagnosed and therefore, untreated until the secondary school level. This may provide clinicians and therapists with alarm indices deriving from the anamnesis and the results of the DTRS that should lead to an earlier diagnosis of reading delay.
- If DTRS was used to create the subsamples, is it not expected that the DTRS predicts pertinence to each sample? Or was it that the author’s goal was determining which DTRS components best predict pertinence?
ANSWER:
Yes, the reviewer is right. Again, we apologize for our mistake. It was not clear in the text and we have clarified it now. As we have already explained above in this reply, for the biomarkers selection algorithm we did not use the DTRS items used for the Boder classification (%of known and unknown words spelled correctly of the spelling test and the reading quotients, the reading level and the reading age). If we use the same items for both procedures, it is clear that our biomarkers algorithm will have a 100% of prediction, since we are giving a very good knowledge to the algorithm. But in this case, we did not include those items in the biomarkers procedure. Therefore, we found a set of biomarkers completely independent from the items used for the Boder classification, which discriminate the two groups with the very high percent of accuracy.
Minor issues
Table information could be condensed (too many tables, too much information per table)
ANSWER:
We decided to remove Table 6, since its information was complementary to Figure 2. Table 7 was renamed as Table 6 instead.
There are some formatting problems, e.g. Ln 20- a dot is missing; Ln 32 – extra space; Ln 64-comma missing; ln 106 – space missing, etc…
ANSWER:
Thank you! We have made a revision of all the text and corrected many errors of this type.
Round 2
Reviewer 1 Report
The authors have now addressed all my concerns.
Still needs attention to the English
Author Response
We thank the reviewer for his/her helpful revision. Another revision of the English has been made through the text, even with the help of the Reviewer 2 and many mistakes have been corrected. Some of the mistakes that remained in the text were corrected working in a clean version of the text. Now we do believe that we have improved the English a lot and the paper is ready to be sent for production.
Reviewer 2 Report
The goals and findings of this study are now clearer, and its relevance is now more obvious to the reader.
I found, however, a significant number of formal problems (I apologize if I didn’t point them before, but some of these came after the edits). I made notes in the submission pdf - attached. I hope these will be helpful. I may have missed some, so I would recommend a very careful check.

Author Response
We deeply thank the reviewer for his/her very careful revision of the text and all the time he/she took for correcting our mistakes and proposing phrases to improve the quality of the writing. We have incorporated them all in the text. We have made an additional reading of the text in the clean version and we have made eliminated redundancies, complex writing and have made the writing more clear and direct. We have removed all the extra spaces and improved the punctuation. We are now happy with the text. We have also made all the references compatible and have changed them to the numbering system suggested by the journal. Now all the references are cited by their number and they are numerated in the reference section.
It has been a very constructive experience working with this reviewer.